# Exploring, measuring and enhancing the coproduction of health and well-being at the national, regional and local levels through comparative case studies in Sweden and England: the 'Samskapa' research programme protocol

Sofia Kjellström,[1] Kristina Areskoug-Josefsson,[1] Boel Andersson Gäre,[1] Ann-Christine Andersson,[1] Marlene Ockander,[1] Jacob Käll,[2] Jane McGrath,[3] Sara Donetto,[4] Glenn Robert[1,4]

¹The Jönköping Academy for Improvement of Health and Welfare, Jönköping University, Jönköping, Sweden
²Djursdala samhällsförening, Djursdala, Sweden
³We Coproduce, London, UK
⁴Florence Nightingale Faculty of Nursing, Midwifery & Palliative Care, King's College London, London, UK

**Correspondence to**
Professor Sofia Kjellström;
sofia.kjellstrom@ju.se

## ABSTRACT

**Introduction** Cocreation, coproduction and codesign are advocated as effective ways of involving citizens in the design, management, provision and evaluation of health and social care services. Although numerous case studies describe the nature and level of coproduction in individual projects, there remain three significant gaps in the evidence base: (1) measures of coproduction processes and their outcomes, (2) mechanisms that enable inclusivity and reciprocity and (3) management systems and styles. By focusing on these issues, we aim to explore, enhance and measure the value of coproduction for improving the health and well-being of citizens.

**Methods and analysis** Nine ongoing coproduction projects form the core of an interactive research programme ('Samskapa') during a 6-year period (2019–2024). Six of these will take place in Sweden and three will be undertaken in England to enable knowledge exchange and cross-cultural comparison. The programme has a longitudinal case study design using both qualitative and quantitative methods. Cross-case analysis and a sensemaking process will generate relevant lessons both for those participating in the projects and researchers. Based on the findings, we will develop explanatory models and other outputs to increase the sustained value (and values) of future coproduction initiatives in these sectors.

**Ethics and dissemination** All necessary ethical approvals will be obtained from the regional Ethical Board in Sweden and from relevant authorities in England. All data and personal data will be handled in accordance with General Data Protection Regulations. Given the interactive nature of the research programme, knowledge dissemination to participants and stakeholders in the nine projects will be ongoing throughout the 6 years. External workshops—facilitated in collaboration with participating case studies and citizens—both during and at the end of the programme will provide an additional dissemination mechanism and involve health and social care practitioners, policymakers and third-sector organisations.

## Strengths and limitations of this study

► Moving beyond the study of individual coproduction projects and taking a longitudinal, multilevel, cross-case approach to explore the complexities of enabling new forms of relationships to improve health and well-being.
► A research group from different disciplines and professional backgrounds including interdisciplinary social science, nursing, medical anthropology, medicine, rehabilitation and sociology.
► A close partnership with practitioners and patients/citizens through an interactive research approach will help coproduce and coevaluate the programme itself.
► A potential limitation is the sociocultural and linguistic differences between the two countries in which the fieldwork will take place.

## INTRODUCTION

The Swedish health and social care system is struggling—as in many other countries—to balance contemporary challenges of increasing demands and rising costs resulting from demographic changes with the opportunities afforded by technological and scientific advances. Within health system redesign and quality improvement internationally, there is currently a 'preoccupation with methods for citizen engagement, public participation and involvement of people with lived experience … Participation has become a distinct cultural and political movement characterised by user involvement in health and social care'.[1] The forms of such involvement span shared decision-making,[2] person-centred care and management at the individual

patient level through quality improvement,[3] research and evaluation at the service and organisational levels[4] and to policymaking at the system level.[5]

The term 'coproduction' is increasingly being applied by those working in the health and social care sectors to refer to any of these forms of collaboration between users and providers of services. Nonetheless, principles underpinning the original conceptualisation of coproduction—such as mutuality and reciprocity—offer the possibility of fundamentally challenging and changing predominant ways of thinking by moving from focusing solely on the delivery of healthcare and social care and towards cocreating health and well-being. This is evidenced in successful, long-standing coproduction projects including, for example, the cocreation of a self-haemodialysis service, as undertaken over several years in the Region Jönköping County in Sweden,[6] through the work of third-sector organisations like 'We Coproduce' in England[7], and by the development and adaptation internationally of approaches such as experience-based codesign.[8]

There has recently been renewed academic interest in—and advocacy for—adoption of coproduction as a means of cocreating value across the public sector.[9 10] In the healthcare context, coproduction is promoted as harnessing the knowledge of patients, carers and staff to make changes about which they care most. It is claimed that 'bringing people together to re-design and improve services as co-producers is re-creating the ways in which public governance, policy and health services are enacted and function'.[1]

### Origins and evolution of coproduction
Originating in the early 1970s, the term coproduction refers to how citizens themselves play an important role in determining the form, delivery and value of public goods and services[11 12]; academic studies of the time were a response to what was seen as a lack of recognition of the role of service users in determining the relative effectiveness of service delivery in different local contexts.[13 14] The creation of time banks—a reciprocity-based work trading system in which hours are the currency where individuals can trade hours of work without paying or being paid for services[15 16]—led to evidence of how collaborative interventions that involve people with long-term psychosocial needs could contribute to improved community links.[17]

Academic interest in coproduction in different sectors has waxed and waned over the subsequent 50 years but the concept is currently attracting greater interest in public management practices generally,[18] including in the healthcare sector specifically. Much of this interest is based on claims of better outcomes and/or efficiency arguments despite empirical evidence remaining limited.[19] With contemporary practices of public sector service delivery highlighting—and espousing the benefits of—efforts to enable coproduction has come renewed critical interest from a range of academic disciplines. As summarised by Palmer et al,[1] commentators have argued that while such interest may reflect a genuine desire to engage citizens in democratic processes, governance and decision-making,[20] it can also be variously perceived as a means to harness citizen efforts and resources as a replacement for reduced public funding (the 'dark side' of coproduction),[21] as a representation of a loss of public value and trust in public sector services and/or a drive to reinvigorate voluntary participation and strengthen community cohesion in response to increasing societal fragmentation.[22] In the healthcare sector specifically, it has been argued that increasing interest in efforts to enable coproduction remain uncritical, lacking acknowledgement of the ethical complexities embedded in welfare (and service) relationships.[22] Others have warned against the risk of the term itself losing meaning as it enters mainstream management discourse and practice, thereby losing association with its radical roots.[23] For the purposes of this research programme, we are following Osborne et al's definition of coproduction as 'the voluntary or involuntary involvement of public service users in any of the design, management, delivery and/or evaluation of public services'.[24]

### Coproduction in the health and social care sector
Notwithstanding definitional issues, Palmer et al argue that we are witnessing 'a political and socio-cultural mindset shift from 'experts know and decide everything' to 'we need to decide things together''.[1] In the context of health services, people are increasingly characterised as designers, learners and actors who can take responsibility for their own health and shape the outcomes that they desire from organisations.[8–10] An excerpt from a recent manifesto for a third (and moral) era in medicine and healthcare is typical of this shift: coproduction, 'codesign' and person-centred care are among the new watchwords, and professionals and those who train them, should master those ideas and embrace the transfer of control over people's lives to the people.[25] While such contemporary interest in coproduction from leaders of quality improvement and improvement science in the health and social care context is a relatively recent development,[26 27] advocates of the need to study and learn from coproduction across the public sector have been doing so for some time.[16 28] A recent Swedish Patient Act supports these trends[29]—as does the latest strategic plan for the National Health Service in England[30]—but there is also a concern that coproduction may become a much used but ultimately meaningless term that everyone says that they can and want to do but without understanding its origins or how to practice and evaluate it.

### Knowledge gaps
We highlight below three significant and interrelated knowledge gaps pertaining to coproduction in the health and social care sectors which our research programme seeks to address: outcomes (measurement), power, power relations and representation (mechanisms) and leadership (management).

## Measurement

Voorberg *et al* conducted a systematic review of the literature (1987–2013) relating to cocreation/coproduction with citizens in public innovation across all sectors.[18] Most studies focused on the identification of influential factors, while hardly any attention was paid to outcomes. Similar conclusions have been drawn by others.[19] Clarke *et al* systematically reviewed outcomes associated with developing and implementing coproduced interventions in acute healthcare settings; there was a lack of rigorous effectiveness and cost-effectiveness studies at both the service and system levels.[23] Greenhalgh *et al* report a narrative review of different models of cocreation relevant to community-based health services; they identify key success principles, such as system thinking, processes of cocreation and leadership styles, but note that 'impact is by no means guaranteed'.[31]

## Mechanisms

Academic reviews of coproduction highlight—with some exceptions[32]—little critical engagement with issues of power, power relations and representation and whether typical patterns of participation serves to make services more or less inclusive (or do they simply reinforce existing social inequalities?).[9 18] In a recent commentary, Batalden—while arguing that where healthcare activities are coproduced, services, providers and service users become far more effective agents of change—notes that current systems can both support and constrain partnerships between patients and professionals (and that historically this kind of partnership has been unequal).[26]

## Management

Management or leadership of coproduction activities should be characterised by the involvement of many stakeholders sharing different perspectives on the same issue. And yet, despite this being described as a founding principle, this is a neglected area as studies of leadership in coproduction initiatives are sparse.[33–35 34 35] A recent review of coproduction initiatives in the UK suggests four main challenges which require negotiation through different styles of leadership[33]:

1. Setting priorities of coproduction and clarifying goals.
2. Guaranteeing greater inclusion of vulnerable and disadvantaged populations.
3. Fostering communication and public accountability.
4. Encouraging and supporting innovative practices and cultural changes that move away from traditional challenges of risk aversion.

The review suggested that the leadership of coproduction initiatives involves several practices over time, emerges as a complex and collective activity (rather than relying on individual leaders) and likely requires a facilitative leadership system and style.[33]

In summary, despite a recent resurgence of interest in the potential of cocreation and coproduction as means not only of maintaining but also improving health and well-being in Sweden and England, there remain significant questions regarding how to measure the impacts of such approaches, the mechanisms by which they achieve those impacts and how they can and should be led.

## AIM

The overall aim of the Samskapa research programme is to explore, enhance and measure the value of coproduction for improving the health and well-being of citizens. Our four research objectives are:

1. To develop, test and establish robust measures of coproduction processes and of the outcomes of coproduction with participants and wider constituencies at different system levels (measurements).
2. To study the social processes and organisational forms that enable inclusive and reciprocal coproduction across and beyond the health and social care sectors (mechanisms).
3. To explore the features of effective systems and styles of leadership that are necessary to enable coproduction (management).
4. To develop explanatory models and other outputs based on a synthesis of existing evidence and analysis of our empirical findings in relation to 1–3 above in order to help enhance the nine participating projects and future coproduction initiatives (model).

## METHODS AND ANALYSIS

This is a 6-year, interactive research programme[36] beginning in 2019 that will provide an overarching platform for mixed-method evaluations of at least nine case studies of coproduction in the health and social care sectors (see table 1 for details of the aims and methods of each of the case studies). The interactive research approach—a form of participatory action research—will place emphasis on joint learning between the participants and the researchers throughout the entire research process, from the definition of the issues to the analysis and dissemination of findings (see figure 1).[36 37] Cross-case analysis and a sensemaking process with participants—how they interpret and give meaning to their experiences[38]—and researchers from the case studies will lead to practical lessons and outputs to assist practitioners, as well as helping address the key research gaps in the current evidence base as identified above: measurement, mechanisms and management.

As well as responding to recent calls for a 'more multidisciplinary framework, using social-psychological, organizational and institutional theories' to form the basis for future coproduction research,[9] the programme has a longitudinal design and will use both qualitative (semistructured interviews, non-participant observation, focus groups) and quantitative methods (eg, patient-reported outcome measures and patient-reported experience measures within feasibility studies of codesigned interventions) (see table 1). We will seek to establish the impact of coproduction, as well as exploring the complexities of

**Table 1** Initial nine case studies (in Sweden unless stated)

| Case study | Project background | Project objective | Research aim | Research methods |
|---|---|---|---|---|
| The Esther network | The Esther network—coordinated by Region Jönköping County—recently won the ICT-Enabled Social Innovation (IESI Award) from the EU Science Hub for best initiative supporting active and healthy ageing. The award was for its positive contribution to society as well as its disruptive ICT-enabled social innovation potential and high level of service integration. From its origins in 1997, the Esther model has subsequently been adopted and implemented in England, Scotland and Singapore. | To improve patient flow and coordination of care. | The story of Esther(s) is a central feature of the Esther Network. In exploring the mechanisms of coproduction in the Network we will focus on the role of narrative. The case study will also provide interesting data on leadership processes. | Qualitative study incorporating documentary analysis; interviews with project leaders and participants; and non-participant observation of network events. |
| Patient Compact | At the regional level, an ongoing strategic innovation programme in Region Jönköping County (the 'Together' programme) started in 2012 and has developed and expanded over time. The programme is divided into several subprojects, of which—what has become—the national development and implementation of a 'Patient Compact' is one. | To transform healthcare delivery closer to citizens, from hospitals to primary care, from primary care to home care and with a focus on health promotion efforts together with other community actors and citizens themselves. | To enhance emerging understandings of coproduction as they evolve over time within the Together programme and establish measures relating to coproduction and its potential impact on health and well-being. | Mixed-methods evaluation incorporating interviews with patients and staff participants at micro, meso and macro levels; documentary analysis; and participant observation of programme events. Secondary analyses of datasets collected as part of programme (eg, clinical outcomes, population health). |
| We Coproduce (UK) | The origins of We Coproduce as a social enterprise in 2013 are rooted in the recognition, at a mental health hospital in London, that service user involvement was not working, with subsequent development of an independent social enterprise. We Coproduce then also began to work with other community organisations and providers across London to help them embed coproduction in their service design and delivery. | Ongoing projects include coproducing a community owned and run radio station; coproducing with frontline mental healthcare staff to support them to make their own films about trauma-based approaches; and coproducing with a local council to embed micro businesses in partnership with bigger local businesses to challenge isolation. | Our research with We Coproduce will focus on exploring the challenges and opportunities of coproducing the implementation of coproduced service standards in mental health wards. | Qualitative study incorporating documentary analysis; interviews with project leaders and participants; and non-participant observation of coproduction events and meetings. |

**Table 1** Continued

| Case study | Project background | Project objective | Research aim | Research methods |
|---|---|---|---|---|
| Djursdala community project | This case study is funded by an European Union initiative and seeks to identify needs and initiate the development and use of digital solutions that promote the health and well-being of a population of ~400 citizens in a rural area. Staff at Jönköping Academy are coordinating research into this initiative which is led by the local community. | To support rural development projects initiated at the local level in order to revitalise rural areas and enhance local community/rural area. | To explore how user-driven digital development can enable cocreated and coproduced services that lead to value for a rural area, and whether digital solutions contribute to sustainable development and, if so, in what way. | To explore the process of using participatory action research to coproduce methods and solutions with local people from the area, through interviews with community leaders, participants and researchers; and participant observation of community-led events. |
| Chemotherapy-induced peripheral neuropathy (UK) | Some cancer drugs cause damage to nerves, a condition called chemotherapy-induced peripheral neuropathy (CIPN). The most common symptoms, felt mainly on hands and feet, are numbness, tingling, pain, muscle weakness and/or sensitivity to cold. People with CIPN can have functional difficulties in carrying out tasks involving their hands and feet. It is important to prepare patients about the possibility of developing CIPN to help them recognise and report symptoms early so healthcare professionals (HCPs) can support them. | To codesign and test an intervention to reduce falls and injuries and improve functional status and quality of life among individuals with CIPN. | To study how codesigned interventions can be developed and put in place early to prevent subsequent CIPN-related falls and injuries, reduce costs to healthcare systems and lessen the burden on HCPs and services. | Feasibility of randomised controlled trial with embedded process evaluation (will include semistructured telephone interviews with all patient participants (n=40) to assess acceptability of the intervention and evaluation methods). Patients will complete outcome measures (early symptom reporting; reduction in symptoms and self-efficacy in managing symptoms; improved functional status; quality of life) at various timepoints. |
| Learning Café: cardiac care | A Learning Café project is underway where people with cardiac care needs come together to collectively discuss how they can improve different aspects of their health and well-being. Clinical measures—as well as patient-reported outcome measures and patient-reported experience measures—are being codesigned with patients and families and professionals. In addition, a codesigned conceptual model of the Learning Café which can be adapted to other groups of patients with chronic disease is being developed. | To explore whether, how and why the coproduction of healthcare services, particularly for individuals (and their families) with cardiac care needs, can contribute to high quality care. | To explore what role motivation plays for patients, family members and healthcare professionals when coproducing healthcare. | Mixed-methods study incorporating patient surveys (sense of security in everyday life, patient satisfaction); focus groups and semistructured interviews with patients and professionals; patient diaries; and non-participant observation. |

Continued

**Table 1** Continued

| Case study | Project background | Project objective | Research aim | Research methods |
|---|---|---|---|---|
| Disabled children and adolescents | This study is mainly taking place in a not-profit organisation in Solberga By, near Stockholm, and is drawing on an action research design to study local quality improvement initiatives to enhance individual support to children with intellectual disability living in special care residence. This includes studying if and how the children's role as coproducers is reinforced by these initiatives. | To enhance staff capacity to design, test and follow-up individual support to children with intellectual disabilities. | The overall aim is to explore the usefulness of integrating improvement knowledge and the International Classification of Functioning, Disability and Health in staff working procedures to improve goal fulfilment and coproduction for children living in special care residence. | Realistic evaluation study design including data collection from observations, behaviour and function assessments, field notes from staff sessions, QI-documentation and focus groups. |
| Therapeutic engagement on an acute psychiatric ward (UK) | Therapeutic engagement has long been regarded as the essence of mental health nursing. Its benefits are well documented: inpatients who are socially engaged adjust better to community life, have greater symptom improvements during treatment and exhibit fewer violent and aggressive behaviours. Nurses who spend more therapeutic time with patients have greater job satisfaction and take fewer sick days, which may reduce the costly use of unfamiliar agency nurses. Despite this, research spanning 35 years shows that just 4%–12% of nurses' time was spent on therapeutic activities. | To empower a service-user group to take a lead role, and in partnership with NHS staff, codesign and implement an intervention to improve nurse–patient therapeutic engagement on acute mental health wards. | To assess the project in terms of improvements in the amount, type and quality of nurse–patient engagement; improved service user/service provider relations; and the fostering of a culture of collaborative working/research practices within a psychiatric ward. | Mixed-methods evaluation incorporating interviews with patients and staff participants; non-participant observation of codesign events; event questionnaires; and a pre–post test design on an intervention and control ward using structured qualitative and quantitative observations, a self-report measure and data from ward registers to assess type, quality and amount of engagement. |
| Learning health system for severe mental illness | In the department for psychosis at Sahlgrenska University Hospital in Gothenburg, a Learning Health System has begun to be developed and tested along with patients, case managers and the management team. Patients and families are active participants in considering system design, user-experience design, choice of outcome measures and development of care processes. | To enable learning throughout the whole 'system' and continuous improvement. | To explore the role of patients in, first, the development of data-visualisation-design and how this impacts on learning both for the patient and their case manager, and second in evaluating outcome measures useful for the patient and their case manager in ongoing treatment. | Mixed-methods study incorporating surveys; interviews; and non-participant observation. |

delivering and improving health and social care through new forms of relationships and partnerships. Through the interactive research process during the course of the 6-year programme, we will be feeding back our emerging findings to the participating case studies as a means of seeking to enhance the impact of their work (figure 1).[39]

**Study setting**

Fieldwork will be conducted in the context of health and social care provision in Sweden and England. The case studies in Sweden will be undertaken in Region Jönköping County, Kalmas, Stockholm and Gothenburg; the English case studies are in London.

**A narrative literature review**

We will begin the programme by conducting a systematic scoping review of the literature. Grant and Booth state that a scoping review is a preliminary assessment of the potential size and scope of existing evidence with the aims to identify the nature and extent of research.[40]

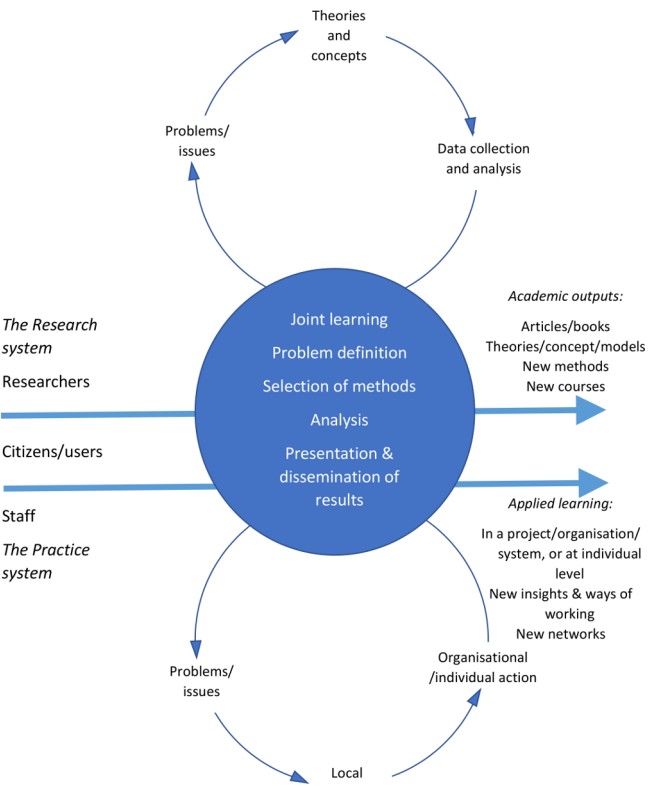

**Figure 1** Interactive research. Adapted from Svensson *et al*[36] and Ellström.[39] Reproduced with permission of Per-Erik Ellström and Budrich UniPress.

Our aims will be to explore current knowledge about coproducing health and well-being, establish best practice within the health and social care sectors and inform the Samskapa programme research objectives by reviewing:

1. How measures relating to participating in coproduction processes and the outcomes of those processes have been developed, by whom and how they have been tested and applied.
2. The mechanisms (eg, social processes and organisational forms) that enable inclusive and reciprocal coproduction.
3. Different individual and collective leadership and management beliefs and practices that enable the coproduction of health and well-being.
4. Which and how explanatory models have been developed, tested and applied with the aim of enhancing coproduction processes and outcomes.

The databases selected for searching are the Cochrane database of systematic reviews, CINAHL, PsycINFO, Medline, PubMed and Scopus. Our inclusion criteria will be peer reviewed, English-language articles that explicitly relate to coproduction or codesign in the health or social welfare context. The findings from the scoping review will directly inform the empirical fieldwork to be undertaken both within and across the nine case studies that together comprise the overall research programme.

## The case studies

Nine case studies of coproduction will form the basis of our empirical fieldwork (table 1). This set of case studies is an opportunistic sample drawn from coproduction projects which members of the research team were either studying—or involved in planning—at the time of our research application; six of the case studies are the subject of planned or ongoing doctoral studies in Sweden or England. There is scope during the 6 years of the programme to purposively add further case studies as they emerge and in order to help meet our four research objectives. The coproduction efforts and processes in each case will be explored by documentary analysis (eg, reviewing key documents such as project protocols and interim reports) as well as qualitative and quantitative methods as relevant to the objectives of each case study (see table 1 for details).

In the following sections, we outline how our scoping review and empirical fieldwork in the nine case studies will combine to inform our four research objectives.

## Measurement

The scoping review (see above) will explore how measures relating to the different forms of value of participating in coproduction processes and the outcomes of those processes have been developed, by whom and how they have been tested and applied (eg, Durose *et al*[41]). We will explore the usefulness of existing measures in the case studies by observing their use (if any) in practice and facilitating interactive workshops with case study leaders on this topic (particularly on how measures and coevaluation of processes and outcomes are integrated in the cases). The precise nature of the observational fieldwork to be undertaken will vary by case study. In most of the cases, this will be through non-participant observation of key meetings and events (and will incorporate any discussions relating to measures). In a minority of cases, more structured forms of observation will be conducted and/or participant observation will be used. We will then codesign with stakeholders and users of the relevant services new generic measures of coproduction (through an interactive process partly informed by the scoping review findings) and then test them over time in the case studies to assess their pragmatic usefulness and generalisability.

How—and to what extent—client/patient involvement is present and acted on in the design and evaluation of outcome measures, as well as the sustainability and understanding of the value of the outcomes, will also be qualitatively explored in the case studies. We will consider the gender and sociodemographic differences (eg, age, gender) among those who actively choose to participate in the coproduction case studies as this may influence both the chosen measurements and the outcomes.[42] The programme will provide an increased understanding of how to develop and use measures within—as well as to evaluate the outcomes of—coproduction projects by illuminating how specific processes in coproduction relate to measured outcomes (see Mechanisms section).[42]

## Mechanisms

We will study the concepts and practices of coproduction to identify mechanisms (eg, social processes and organisational forms) that contribute to or hinder the development of values and actions that enable inclusive and reciprocal coproduction. First, our scoping review (see above) will identify previous studies (eg, Palmer *et al*[1]) which have sought to explore such mechanisms. The review will establish what is already known about key issues such as power and levels of representativeness, and any interventions or modifications that have attempted to resolve these. Second, informed by the narrative review findings, we will then draw on our emerging empirical findings relating to the local conceptualisations and practices in the nine case studies to design complementary or alternative ways of working. We will subsequently observe the implementation of these in one or more of the case studies to assess whether and how they enhance the coproduction efforts.

## Management/leadership

Our scoping review will establish what is known about different individual and collective leadership and management assumptions that enable the coproduction of health and well-being. Leadership will be framed as complex, interactive and dynamic, and analysed in a way that contributes to generative learning and theoretical transferability.[43 44] Studying leadership needs to shift away from a focus on leaders and followers styles and towards a system of processes through which a collective endeavour unfolds. The sensemaking and assumptions about leaders and leadership has a central role in how leadership is enacted and can be evaluated, developed and influenced.[45] This approach builds on a belief that individual and collective assumptions about leadership shape how team organisational members work with leadership within an organisation. Our longitudinal study design will enable us to address whether leadership assumptions vary across cases, how they change and develop over time and how people in different positions of power participate in various leadership activities and sensemaking processes?[45 46] We will explore leadership across the nine case studies through a range of methods including semistructured interviews, surveys and non-participant observation of meetings and events. Workshops will also be facilitated with researchers and key actors to elicit their views on how leadership has been enacted in each of the cases. The results will be summarised in a dynamic system assessment tool that seeks to aid understanding of how to analyse and improve coproduction leadership.

## Models

Based on the multilevel and longitudinal case studies in different health and social care settings in Sweden and England, we will conduct a meta-synthesis.[46–50] We will use theoretical perspectives—identified and selected through our scoping review (see above)—and systems thinking to make sense of the context and cases.[43 51] In the meta-synthesis, we will explore patterns in both the qualitative and quantitative data from across the different case studies and examine and compare differences and similarities across these. To inform this cross-case analysis—in addition to the 'within' case fieldwork outlined in table 1—longitudinal semistructured interviews will be conducted at least annually during the 6 years of the programme with key leaders of each of the nine case studies. These will focus on our four objectives—measurement, mechanisms, management and models—and how practices of co-production are being enacted over time. We will synthesise the findings from the within and 'across' case study analyses to identify relevant themes, similarities and differences between the cases (figure 2).

This work will develop explanatory models for successful coproduction through analysis of all cases and synthesis of our findings in relation to measurement, mechanisms and management. Further outputs will include the code-sign of tools to enable those participating in coproduction projects to reflect on the maturity of their efforts and how their work might be enhanced. The narrative review will establish which and how explanatory models have been developed, tested and applied with the aim of enhancing coproduction processes and outcomes. As a part of our interactive approach, we will convene and facilitate multiple stakeholders meetings (with leaders, participants and researchers of the nine case study projects) in the format of Joint Interpretive Forums (JIFs)—a form of group discussion which aims to foster 'perspective taking' and joint decision-making—to enable the collaborative interpretation of both our review and empirical research findings and the development of actionable recommendations for policy and practice.[52] The first JIF will be convened at the end of year 3; a JIF will then be held at 6-month intervals for the remaining duration of the programme (7 JIFs in total). The initial and final JIF will be open to all participants while the others will focus on specific aspects of the emerging model(s) and invitees will be selected as appropriate. Informed by the findings from above all the JIFs will be held in Sweden. Importantly, all PhD students who are studying one of the case studies will be integral members of this ongoing sensemaking process giving them further personal development opportunities.

## Patient and public involvement

Jönköping Academy for Improvement of Health and Welfare (JA) is the host of the research programme and has an established tradition of interactive research where knowledge is created in the interaction between theory and practice. 'Practice' has traditionally been represented by professionals in this model but in this programme we will use and expand this to include citizens and patients as full partners (see figure 1). Processes to enhance partnership working through ongoing, joint design of the specific research materials and methods to be applied in each of the case studies will therefore be both a result and a phenomenon for study in the programme. As part

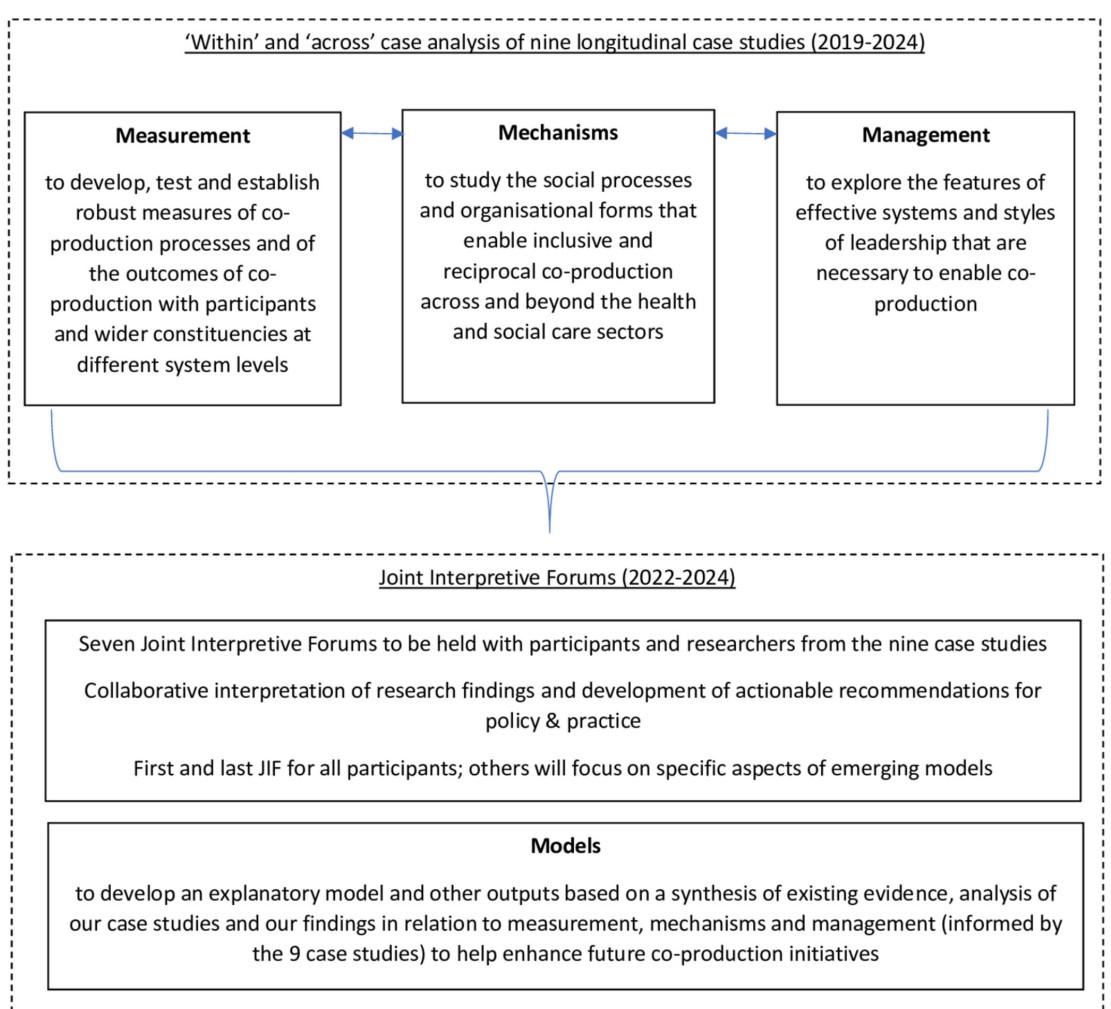

**Figure 2** Cross-case analyses: measurement, mechanisms, management and models.

of our ongoing interactive approach, members of the research team have identified several 'charters for coproduction' (tools to aid reflective practice within co-production projects) from the UK and the USA and had these translated into Swedish. These will form the basis for reflective dialogues which will test the appropriateness of applying these various materials in the Swedish context.

Through conversations with public and third-sector organisations already engaged in coproduction in health and social care, the Region Jönköping County in 2016 decided to financially support development of an International Centre for Coproduction hosted by JA as a sister Centre to a similar Centre at the Dartmouth Institute, New Hampshire in the United States. In 2017, Jönköping University made a strategic decision to support the research capacity in the Centre at JA through investing in senior research positions in coproduction and a project leader. In addition, trade unions and small and medium size enterprises have been involved in ongoing discussions. Processes to integrate user and public representatives are in progress and they can contribute in several ways; in the codesign of service innovations but also in interactive research processes to assure the relevance of questions and the validity of results. The centre provides infrastructure that supports the involvement of patients and citizens in research processes as well as in practice. From the UK, a community-based organisation We Coproduce will form one of our case studies and leaders of this organisation will also engage with the doctoral and post-doctoral students in a regular series of practice-based workshops. The leaders of the Djursdala and We Coproduce case studies have contributed to and are coauthors of this protocol.

Partnership working is an inherent feature of each of the nine case studies of coproduction which we will initially be studying. At the programme level, there will be meetings throughout the 6-year programme, with all the current authors, participating post-doctoral researchers and doctoral students, as well as practitioners and citizens from the six Swedish case studies. Through dialogue, cross-case study sharing of knowledge and development of further research questions and thoughts of interest, the members will help codesign, coproduce and coevaluate the programme as a whole; these interactions will be supported by specific seminars. There will be dissemination workshops within and between each of the case studies to enhance cross-case study knowledge creation and networking.

## ETHICS AND DISSEMINATION
### Ethics
Ethical approval will be obtained from the regional Ethical Board in Sweden for all of the case studies where this is applicable and from the relevant authorities in England for the case studies there. All data and personal data will be handled in accordance with the new European General Data Protection Regulation (EU) 2016/679. Given the need to respect the integrity, autonomy and privacy of the participants, it is important to acknowledge that ethical considerations and principles may exert an influence on the research design. Informed consent will be accomplished by mutual communication where the researcher provides accurate information and listens to the individual participants in order to make sure that they comprehend and make voluntary choices to participate, not only at initial recruitment but also throughout participation in the project. This is particularly important for vulnerable groups,[53] some of which will be participating in this programme (see final three projects in table 1). While the interactive research design (ie, including patients, staff and stakeholders in the whole research process) can be challenging to Ethical Boards—posing difficulties in detailing the nature and timing of the research to be undertaken beforehand—the research team have extensive experience of successfully navigating these ethical processes in both Sweden and England.

### Dissemination
Our intention is to engage strategically with five audiences with whom we will deepen and sustain existing—and create new—relationships to help inform our ongoing research and to provide opportunities to create positive change in health and social care services:
1. Leaders in the international, national and regional planning of health and social care services.
2. Educators developing the next generation of health and social care professionals.
3. Professional bodies and trade unions.
4. Citizens as they access, use and shape services.
5. Research funders.

We will adopt a structured approach to mapping key organisations, networks and opinion leaders at international, national and regional levels; this will be part of our work both in terms of developing a communications strategy in year 1 of the programme and to help us identify diverse and influential members of our Advisory Board. As part of this approach, we will engage with national and regional leaders of health and social care services through contact with the 'Swedish Association of Local Authorities and Regions' (SALAR). The Department for Health and Social Care at SALAR has responsibility, for example, for supporting the development of elderly care, social care, public health, disability, quality and safety, and equity. SALAR also coordinates several national networks of leaders from across all Swedish regions and counties providing opportunities to share our findings and outputs to contribute to policy discussions and practice.

A further prime opportunity for dissemination is through involving educators and students—often mid-career professionals—involved with a Master programme in 'Leadership for Improvement of Health and Welfare' developed and run at the Jönköping Academy, Jönköping University; this will enable us to engage with students involved in leading improvement of care throughout Sweden. We also have connections with national bodies with an interest in integrating knowledge on coproduction into different levels of education. A process of interaction with professional bodies and trade unions started in March 2018 and will enable further collaboration during the research programme. Citizens are both key participants in—and an important audience for—the programme and will be invited to participate and integrated throughout our work; we will also approach Swedish patient organisations. While there is no national umbrella patient organisation, there are many disease/condition-specific patient advocacy groups which offer opportunities for testing and scaling up coproduction initiatives. Finally, contact has also been initiated by the funder with those leading parallel research programmes on coproduction taking place in Sweden raising the potential for increasing adoption of the findings within and across programmes.

In combination with the interactive research approach outlined above, the proposed involvement of researchers, practitioners and citizens will enable fruitful ways of dissemination and impact throughout the 6-year Samskapa programme, enhancing the likelihood that co-production can be a catalyst for new forms of relationships to deliver and improve health and well-being.

**Acknowledgements** We are grateful to the following doctoral students for participating in the Samskapa research programme and providing details of their studies as presented in table 1: Andreas Gremyr (Jönköping University), Sarah McAllister (King's College London), Sofia Persson (Jönköping University), Anne-Marie Suutari (Jönköping University), Mary Tanay (King's College London) and Pontus Wallin (Jönköping University).

**Contributors** SK is principal investigator and a member of the executive group of the research programme. She designed the overall research study, read and contributed to revisions and additions to the manuscript and approved the final version. KA-J is a member of the executive group of the research programme. She read and contributed to revisions and additions to the manuscript and approved the final version. BAG is a member of the executive group of the research programme. She read and contributed to revisions and additions to the manuscript and approved the final version. A-CA, MO, SD, JK and JM read and contributed to revisions and additions to the manuscript and approved the final version. GR is a member of the executive group of the research programme. He helped design the overall research study, drafted the original manuscript and contributed to revisions and additions to the manuscript and approved the final version.

**Funding** This work is supported by Forte: Swedish Research Council for Health, Working Life and Welfare, grant number 2018-01431.

**Competing interests** None declared.

**Patient consent for publication** Not required.

**Provenance and peer review** Not commissioned; externally peer reviewed.

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
