## [Reviewer comments · BMJ Open]

ARTICLE DETAILS

TITLE (PROVISIONAL)	Exploring, measuring and enhancing the co-production of health and wellbeing at the national, regional and local levels through comparative case studies in Sweden and England: the 'Samskapa' research programme protocol
AUTHORS	Kjellstrom, Sofia; Areskoug-Josefsson, Kristina; Gäre, Boel Andersson; Andersson, Ann-Christine; Ockander, Marlene; Kall, Jacob; McGrath, Jane; Donetto, Sara; Robert, Glenn

VERSION 1 – REVIEW

REVIEWER	Karen Carlisle James Cook University, Australia
REVIEW RETURNED	04-Mar-2019

GENERAL COMMENTS	A very interesting and timely study. The protocol paper however lacked details which allowed me to fully understand the context (background) and the methods used and would recommend major revisions to the paper. In addition the research objectives require clarification. Below are a number of comments and suggestions: Introduction: First paragraph the first two sentences start with As and htey read quite clumsy so suggest rewording/rephrasing Sentence beginning Line 13 first paragraph, it would be very helpful to the reader if you provide references to examples or frameworks A linking sentence to the second paragraph would improve readability Suggest table 1 is changed to a narrative rather than in the table form it is currently presented. Also in the timeline there is a gap of 5 years, it would be good to know what happened in the intervening years. Section coproduction in the health and social care sector there are references to a Swedish Patient Act. Given the context of the study some reference to the UK would provide balance. In general I would like to see more of the policy context to frame the study. Knowledge gaps section suggest removal of bullet points as the information you are presenting could be written in narrative. This section also had lots of very long sentences that were hard to follow, please review and simplify to improve readability. p5 line 42 " A recent review of leadership literature and the four main challenges needs much more information in order to see the relevance- what was the context of the study? Were there any solutions? Aims p6 lines 10-24. You mention that yu are measuring the value of co-production but it is not clear in the section what you mean by value and what will be collected to determine the value in terms of measuring, exploring and enhancing. For readers it needs to be explicit. In this section it is also not clear whether "enhance" the value of coproduction is applied within the scope of the study i.e. an
---

	intervention. I think that is your intention but it is not very clear here. Methods and analysis (p6 line 28) Overall the methods and analysis require much more detail to understand what is planned. Provide a reference or underpinning for interactive research design - is it Participatory Action Research? How were the case studies opportunistically recruited? Second paragraph in this section p6 line 47 you mention you are seeking to establish and enhance the impact of co-production - please elaborate on what you mean and also make it clear if interventions are part of the study. Figure 1: Is good and I would like to see something more on the intersect of local theories (staff) and theories and concepts. I would also imagine the outputs would be more than academic and usefulness - new ways of working? shared dissemination? p7 before project management section i feel there should be a linking section or some explanation as to why the project management and narrative literature review sections are there. In addition there is not enough detail in either section to understand how they fit with the overall study and how they will be undertaken. The case studies section requires more detail on qualitative and quantitative methods. Presentation of case studies in table 2 requires more consistency on the information on each case study. Figure 2 provides a little clarity but more information is needed and i would recommend a paragraph/preamble to signpost the reader to the following paragraphs on the research objectives. Measurement p10 line 5 more detail on the narrative review and how you will observe the use of existing measures in practice and the interactive workshops. How will you measure client/patient involvement as mentioned in lines 17-27 p 10. Mechanisms - First sentence, it is not clear if this will be from the case studies or narrative review. Also more information on how you will test these (line 40). Management and leadership - some more explanation on how leadership beliefs and practices change and develop over time. Models p11 the term heuristic tool is mentioned for the first time as part of further outputs, please elaborate. Also explain how you will conduct the narrative review to inform modules
--	--

REVIEWER	Brita Roy Yale School of Medicine, USA
REVIEW RETURNED	15-Mar-2019

GENERAL COMMENTS	Overall, this is a very well-written and well-designed protocol paper describing a cross-national study of 9 existing co-production programs in various settings that will address three major gaps in the co-production literature: measurement, mechanisms, and management. Some suggestions to better organize the manuscript and clarify the methods and processes follow: ABSTRACT: Lines 14-19: The "three fundamental issues" are the knowledge gaps this study aims to address. Suggest combining the second and final sentences of this paragraph so that is clear. Line 21: Does "confirmed" mean existing co-production projects? Or confirmed to participate in this study? Might others join later? Lines 35-38: Could be more specific here about the audiences targeted for dissemination.
---

	TITLE Recommend including the 3 knowledge gaps you plan to address in the title. Needs to be more explicit about the aim(s) of the study. INTRODUCTION Overall, could shorten the introduction a bit, and use more space to highlight the knowledge gaps. Page 5, lines 15-18: The sentence about the creation of time banks does not clearly articulate the implications of the study without looking up the references - suggest summarizing the study in a more descriptive way. Page 6: It would be helpful to have subheadings so it's clear where the discussion about each knowledge gap is. Line 20: Typo - "associates" should be "associated" METHODS Page 7, Line 26: some readers may not be familiar with the term "sensemaking." Suggest adding a definition. Line 44: Qualitative methods are stated to be used, but the details regarding the qualitative methods are scarce in this protocol. Suggest adding some detail about the qualitative methods planned. Page 8, Line 9: not sure what "work packages" are. Line 23: It is only now clear that you incorporated 9 existing co-production programs for the case studies in this paper. Would make that more clear earlier. Line 31: Typo - there is a ")" after qualitative that should not be there. Lines 31-32: Unclear what "documentary analysis" refers to. Also, why would the methods vary across case studies? Later, it seems like you would be testing existing or new instruments across all sites. Maybe an example would help, or including the planned method(s) for evaluation for each case study within Table 2. Page 10 Line 40: Consider placing the Cross-case analyses section after describing measures or integrate with Model development ETHICS AND DISSEMINATION Page 13, Line 31: There are several places where it is noted that vulnerable groups will need specific attention in both incorporating into co-production, but also for evaluation. It would be helpful to know if there are specific vulnerable groups that will be included in the study design (perhaps within each case?) and how their experiences will be assessed.
--	--

VERSION 1 – AUTHOR RESPONSE

Reviewer: 1 Karen Carlisle	
A very interesting and timely study. The protocol paper however lacked details which allowed me to fully understand the context (background) and the methods used and would recommend major revisions to the paper. In addition, the research objectives require clarification.	Thank you for your interest in our study. In response to your feedback and comments we have revised the manuscript as detailed below. In particular we have substantially revised the table presenting the 9 case studies and the research methods to be used to study each as we believe this will address several aspects of your feedback.

Introduction:  • First paragraph the first two sentences start with As and they read quite clumsy so suggest rewording/rephrasing • Sentence beginning Line 13 first paragraph, it would be very helpful to the reader if you provide references to examples or frameworks • A linking sentence to the second paragraph would improve readability • Suggest table 1 is changed to a narrative rather than in the table form it is currently presented. Also in the timeline there is a gap of 5 years, it would be good to know what happened in the intervening years. 	 • The first two sentences in the first paragraph have been rephrased • We have now provided references for each of the following: shared decision-making, person-centred care, research & evaluation at the service/organisational level, and policy-making at the system level • We have added a linking sentence between the first and second paragraphs • We have removed table 1 and now provided a reference for the hemodialysis example in the text.
Section coproduction in the health and social care sector there are references to a Swedish Patient Act. Given the context of the study some reference to the UK would provide balance. In general I would like to see more of the policy context to frame the study.	We have now included a relevant reference to the UK policy context.
Knowledge gaps section suggest removal of bullet points as the information you are presenting could be written in narrative. This section also had lots of very long sentences that were hard to follow, please review and simplify to improve readability.	The bullet points have been removed and the information is now in the narrative. We have reviewed and revised this section by amending long sentences to improve readability.
p5 line 42 " A recent review of leadership literature and the four main challenges needs much more information in order to see the relevance- what was the context of the study? Were there any solutions?	We have provided further details of the context and findings of this review as requested.
Aims p6 lines 10-24. You mention that you are measuring the value of co-production but it is not clear in the section what you mean by value and what will be collected to determine the value in terms of measuring, exploring and enhancing. For readers it needs to be explicit. In this section it is also not clear whether "enhance" the value of coproduction is applied within the scope of the study i.e. an intervention. I think that is your intention but it is not very clear here.	We have now provided further clarification of how the scoping review will establish the different forms of 'value' that have previously been ascribed to the co-production of health and welfare, and that these forms of value will be explored in our case studies (in the relevant 'measurement' section on page 12). We have amended the wording of objective 4 to make it clear that we will be feeding back our emerging findings to the nine case study sites (as part of our interactive research approach), and thereby seeking to enhance the value of co-production.

Methods and analysis (p6 line 28) Overall the methods and analysis require much more detail to understand what is planned.	We have now provided further details as to the methods and analysis which will be undertaken both within and across our 9 case studies. The major revisions are:  • we have added a new column to table 1 detailing the specific research methods to be used in each of the case studies (section entitled 'The case studies') • we have provided further detail as to the methods and analysis which will inform our approach to the cross-case analysis (section entitled 'Models') • we have indicated to the reader in the 'Methods and analysis' section that further detail is provided in these two latter sections of the manuscript
Provide a reference or underpinning for interactive research design - is it Participatory Action Research?	Interactive research is a particular extension of Participatory Action Research. We have amended the text and provided a further reference to interactive research (page 7); figure 1 is adapted from two overviews of interactive research (both of which are cited and acknowledged).
How were the case studies opportunistically recruited?	We have amended the text to make it clearer how the 9 case studies were recruited.
Second paragraph in this section p6 line 47 you mention you are seeking to establish and enhance the impact of co-production - please elaborate on what you mean and also make it clear if interventions are part of the study.	We have amended the text at this point to make clear that - through the interactive research process - we will be feeding back on emerging findings during the course of the 6 year research programme to the participating case studies as a means of seeking to enhance the impact of their work.
Figure 1: Is good and I would like to see something more on the intersect of local theories (staff) and theories and concepts. I would also imagine the outputs would be more than academic and usefulness - new ways of working? shared dissemination?	The intersect of 'local theories' (staff) and 'theories and concepts' occurs in the joint learning etc in the centre of figure 1 (i.e. where the research and practice systems are brought together). We had included 'new ways of working' within 'new insights' but have now made this explicit.
p7 before project management section i feel there should be a linking section or some explanation as to why the project management and narrative literature review sections are there. In addition there is not enough detail in either section to understand how they fit will the overall	We have deleted the project management section but provided significantly more detail on the systematic scoping literature review section which is an important aspect of the overall research programme, informing the empirical work to be conducted in and across the nine case studies.

study and how they will be undertaken.	
The case studies section requires more detail on qualitative and quantitative methods.	We have added a new column to table 1 detailing the qualitative and quantitative research methods to be used in each of the nine case studies.
Presentation of case studies in table 2 requires more consistency on the information on each case study.	We have now revised the presentation of the details of each of the case studies in table 1 (formerly table 2) so that the information relating to each case study is more consistent.
Figure 2 provides a little clarity but more information is needed and i would recommend a paragraph/preamble to signpost the reader to the following paragraphs on the research objectives.	We have now provided a preamble to signpost the reader to the subsequent paragraphs relating to each of our four research objectives.
Measurement p10 line 5 more detail on the narrative review How you will observe the use of existing measures in practice and the interactive workshops? How will you measure client/patient involvement as mentioned in lines 17-27 p 10.	We have provided more detail about the systematic scoping review on page 8 and cross refer to this on p.10 (beginning of 'measurement' section) The precise nature of the observational fieldwork to be undertaken will vary by case study. In most of the cases this will be through non-participant observation of key meetings and events (and will incorporate any discussions relating to measures). In a minority of cases more structured forms of observation will be conducted and/or participant observation will be used. The final column in table 1 distinguishes between these different forms of observational fieldwork. We have provided this clarification at this point in the text. We will not formally 'measure' client/patient involvement in a quantitative sense but rather qualitatively explore 'How - and to what extent - client/patient involvement is present and acted upon in the design and evaluation of outcome measures, as well as the sustainability and understanding of the value of the outcomes'. We have made this clearer in the relevant text.
Mechanisms - First sentence, it is not clear if this will be from the case studies or narrative review. Also more information on how you will test these (line 40).	We have now clarified that both the systematic scoping review and our empirical fieldwork will inform our consideration of mechanisms (p.12-13). We have amended the wording at this point in the manuscript to 'We will subsequently observe the implementation of these in one or more of the case studies to assess whether and how they enhance the co-production efforts' (as 'test'

	implied we already have a view as to likely evaluation approach which is not the case; this will emerge during the 6 year programme and is not pre-defined).
Management and leadership - some more explanation on how leadership beliefs and practices change and develop over time.	We have now revised and expanded the 'Management and leadership' section on p.13.
Models p11 the term heuristic tool is mentioned for the first time as part of further outputs, please elaborate. Also explain how you will conduct the narrative review to inform modules	We have clarified what we intended by the phrase 'heuristic tool' by revising the text. As noted above, we have now provided greater details of our systematic scoping review earlier in the manuscript (page 8).
Reviewer: 2 Brita Roy Overall, this is a very well-written and well-designed protocol paper describing a cross-national study of 9 existing co-production programs in various settings that will address three major gaps in the co-production literature: measurement, mechanisms, and management.	Thank you for your comments.
ABSTRACT: Lines 14-19: The "three fundamental issues" are the knowledge gaps this study aims to address. Suggest combining the second and final sentences of this paragraph so that is clear.	We have now amended this section of the abstract in line with the suggestion.
ABSTRACT: Line 21: Does "confirmed" mean existing co-production projects? Or confirmed to participate in this study? Might others join later?	We have replaced 'confirmed' with 'ongoing' to make this clearer.
ABSTRACT: Lines 35-38: Could be more specific here about the audiences targeted for dissemination.	We have amended this section of the abstract to make this clearer and more specific.
TITLE: Recommend including the 3 knowledge gaps you plan to address in the title. Needs to be more explicit about the aim(s) of the study.	We have considered this proposal but do not feel this is necessary. However, if the editor agrees with this suggestion from the reviewer then we can provide an amended title.
INTRODUCTION: Overall, could shorten the introduction a bit, and use more space to highlight the knowledge gaps.	We have shortened the introduction in line with comments also from reviewer 1 and provided more details regarding each of the knowledge gaps.
INTRODUCTION: Page 5, lines 15-18: The sentence about the creation of time banks does	We have amended this section of the text to summarise the study in a more descriptive way.

not clearly articulate the implications of the study without looking up the references - suggest summarizing the study in a more descriptive way.	
INTRODUCTION: Page 6: It would be helpful to have subheadings so it's clear where the discussion about each knowledge gap is.	We have now added sub-headings to each of the knowledge gap sections.
INTRODUCTION: Line 20: Typo - "associates" should be "associated"	Amended.
METHODS: Page 7, Line 26: some readers may not be familiar with the term "sensemaking." Suggest adding a definition.	We have clarified our use of the term 'sensemaking' here.
METHODS: Line 44: Qualitative methods are stated to be used, but the details regarding the qualitative methods are scarce in this protocol. Suggest adding some detail about the qualitative methods planned.	In response also to reviewer 1 we have now provided more details about the methods to be used in each of the nine case studies (see additional column in table 1). In the 'Methods and analysis' section we have now provided more detail as well as referring the reader to table 1.
METHODS: Page 8, Line 9: not sure what "work packages" are.	We have now deleted the 'project management' section.
METHODS: Line 23: It is only now clear that you incorporated 9 existing co-production programs for the case studies in this paper. Would make that more clear earlier.	We have stated that there are nine coproduction projects in the research programme in both the abstract and immediately after our aims and research objectives.
METHODS: Line 31: Typo - there is a ")" after qualitative that should not be there.	Amended.
METHODS: Lines 31-32: Unclear what "documentary analysis" refers to. Also, why would the methods vary across case studies? Later, it seems like you would be testing existing or new instruments across all sites. Maybe an example would help, or including the planned method(s) for evaluation for each case study within Table 2.	As stated above – and also in response to similar comments from reviewer 1 – we have now provided further details about the methods to be used in each of the nine case studies in our substantially revised table 1. With regard to the specific points raised:  documentary analysis will be used in several of the case studies and will include review of key documents relating to these (for example projects protocols and interim reports). We have now included wording to this effect in the 'case studies' section and indicated in table 1 which case studies will include this method the methods vary across the case studies because both the project objectives and

	research aims (contributing to our overall research objectives) relating to each are different, and therefore require different research methods. We hope the substantially revised table 1 now makes this clear  • we have made now clearer in table 1 which case studies will include formal testing of either co-produced/co-designed interventions or new measures
METHODS: Page 10 Line 40: Consider placing the Cross-case analyses section after describing measures or integrate with Model development	Thank you for this suggestion. We have now integrated the 'cross-case analysis' section within the 'Models' section.
ETHICS AND DISSEMINATION: Page 13, Line 31: There are several places where it is noted that vulnerable groups will need specific attention in both incorporating into co-production, but also for evaluation. It would be helpful to know if there are specific vulnerable groups that will be included in the study design (perhaps within each case?) and how their experiences will be assessed.	In the 'Ethics & Dissemination' section we have now cross-referenced to table 1 which makes clear that the final three of the nine case studies are likely to include vulnerable populations. Details of how experiences of vulnerable groups will be assessed are provided in relation to the relevant case studies in table 1.

VERSION 2 – REVIEW

REVIEWER	Karen Carlisle College of Medicine and Dentistry James Cook University Australia
REVIEW RETURNED	13-May-2019
GENERAL COMMENTS	Thank you for the opportunity to review this manuscript again and I am very satisfied that all comments/queries in my first review have been addressed. I have no further comments to add.
REVIEWER	Brita Roy Yale School of Medicine
REVIEW RETURNED	22-May-2019
GENERAL COMMENTS	The authors have sufficiently addressed my comments on the originally submitted version of the manuscript.